# Mutual generation in neuronal activity across the brain via deep neural approach, and its network interpretation

Ryota Nakajima[1], Arata Shirakami[1], Hayato Tsumura[1], Kouki Matsuda[1], Eita Nakamura[2,5] & Masanori Shimono [1,3,4✉]

In the brain, many regions work in a network-like association, yet it is not known how durable these associations are in terms of activity and could survive without structural connections. To assess the association or similarity between brain regions with a generating approach, this study evaluated the similarity of activities of neurons within each region after disconnecting between regions. The "generation" approach here refers to using a multi-layer LSTM (Long Short-Term Memory) model to learn the rules of activity generation in one region and then apply that knowledge to generate activity in other regions. Surprisingly, the results revealed that activity generation from one region to disconnected regions was possible with similar accuracy to generation between the same regions in many cases. Notably, firing rates and synchronization of firing between neuron pairs, often used as neuronal representations, could be reproduced with precision. Additionally, accuracies were associated with the relative angle between brain regions and the strength of the structural connections that initially connected them. This outcome enables us to look into trends governing non-uniformity of the cortex based on the potential to generate informative data and reduces the need for animal experiments.

[1] Kyoto University, Graduate School of Medicine, Kyoto, Japan. [2] Kyoto University, Graduate School of Informatics, Kyoto, Japan. [3] Kyoto University, The Hakubi Center for Advanced Research, Kyoto, Japan. [4] Osaka University, Graduate School of Informatics, Kyoto, Japan. [5] Present address: Kyoto University, The Hakubi Center for Advanced Research, Kyoto, Japan. ✉email: m-shimono@ist.osaka-u.ac.jp

In the nervous system, a large number of neurons are repeatedly firing as they interact with each other. This scene has been likened to a symphony of complex spikes[1–3]. When measuring neural activity, it is customary for electrophysiologists to discern from the sound of spikes whether the measurement points are in contact with active neurons, and then to be satisfied or disheartened. What characteristics does the temporal flow of such neural activity symphonies have?

It is also known that certain long-time correlations exist in the spike time series of neural activity[4–7], indicating that an activity state of a neuron population at a time already has some information about its future state. The state of whether or not an individual neuron fires is essentially maintained as an interdependent relationship among multiple neurons, rather than maintained individually for the inherent activity mode of each neuron. In other words, it is crucial to simultaneously acquire the activity of a large number of neurons to understand their correlation.

The ability to simultaneously record from multiple neurons has markedly improved over the years, thanks to advances in electrode technologies. From the advent of transistor computers and microelectrode probes in the 1950s, there has been a remarkable trend where the number of neurons that can be monitored simultaneously has approximately doubled every seven years[8]. Today, with the advent of novel electrode technologies, we can record activity from hundreds, thousands, or even tens of thousands of neurons at the same time[9–11]. These advances have been fueled in part by improvements in the scalability and accessibility of input/output interfaces, the reduction of electrode sizes to afford a higher density and sampling resolution, and the design of macroporous structures to increase the sampling volume without causing substantial damage to neural tissues[9,10].

Thanks to these advances, we now have the capability afforded by recording technology necessary to reliably verify how accurately we can generate the future activity of individual neurons from past activity of the neuron population.

The nervous system is active even in the absence of external stimuli. Such neural activity is called spontaneous activity. For a long time, neural activity has been measured while animals undertake any tasks and the neural activities have been evaluated in correlation with the task. In fact, more than 80% of the energy in the brain is expended in the task-free state, and spontaneous activity consumes most of the energy of neural activity[12].

As we will discuss later, it is also known that stimulus-induced activity is fundamentally rooted in the state of preceding spontaneous activity.

In the past, when many neurons could not be measured simultaneously, temporal changes in the activity of individual neurons were regarded only as classical stochastic activity. However, recent measurements have shown that spontaneous activity is also considered to retain a causal relationship between activities with a degree of inevitability[13,14].

On a macroscopic (anatomical) scale, spontaneous activity has been observed to produce specific patterns throughout the brain. A typical example is the default mode network, a pattern of activity that is inversely correlated with presentations of external stimuli [Raichle et al.]. It is also clear that there are multiple other modes in the macroscopic spontaneous activity patterns[15].

This massive amount of research on spontaneous activity on a macroscopic scale forms a huge research field that continues to this day. The spontaneous functional activity patterns can also be systematically interpreted by comparing them to structural wiring[16–19].

The measurement and analysis of spontaneous activity of neurons at the microscale have been pursued both in vitro and in vivo. Classically, the firing timings of individual neurons have been quantified as a deviation from the Poisson point process generated when we regard them as a random time series[20,21]. Randomness and simple repetitive patterns have also been assumed in the activity patterns of multiple neurons.

Recent studies have begun to capture the presence of complex, but non-random rules within these patterns. One of pioneering studies, using real-time optical imaging, revealed the patterns of spontaneous activity observed when multiple neuronal activities are measured simultaneously. For example, in the rodent visual cortex, spontaneous activity was found to intrinsically exhibit variations of spatial patterns in evoked activity even before stimulus presentation[22]. The same research team also demonstrated that activity patterns obtained from optical imaging time-locked to the firing timing of single neurons show clear similarity to patterns time-locked to evoked activity[23].

Such interactions between multiple neurons have sequential patterns caused by a series of inevitable interactions that are thought to be mediated by synaptic connections between neurons.

The connections of quantified causal interaction between neurons drawn as arrows are called effective connectivity. Much work has also been done to reconstruct structural wiring as effective connectivity reconstructed from neural activity[24–26]. It has also been pointed out that networks reconstructed from neural activity are closely related to stimulus-dependent evoked activity of neurons[27].

When we consider a symphony of neural activity (i.e., a coordinated flow in time) as music, we notice an interesting technical diversion. Polyphonic music includes multiple musical notes sounding simultaneously as found in piano music and ensemble music which can be regarded as time series data similar to multicellular spikes data. We are mapping the pitches of musical notes to the neurons and the onset time to the time of firing. The existence of co-occurrence relationships and long-time correlations between specific pitches is similar for music data.

Attempts to automatically generate music have been made since the 1950s[28]. Many have also utilized Artificial Neural Networks (ANNs) for music generation[29–32]. Along with recent advances in data analysis techniques using ANNs, music generation techniques based on ANNs have been notably improved. We can expect that it is also possible to generate spike data with properties similar to those of real-world music.

Given the similarities between artificial neural networks (ANNs) and motifs in the nervous system, it might be also expected that ANNs could be applied to generate spike data with properties resembling real-world neural activity[33]. There were attempts to analyze neural spike data can be traced back to as early as the 1960s, with seminal work by pioneers such as Wilfrid Rall, whose mathematical models used differential equations to describe the temporal dynamics of neuronal electrical activity. Rall's work was essential in laying the groundwork for what would eventually become spiking neural network (SNN) models. Another notable contributor to this field is Carver Mead, who made substantial strides in neuromorphic computing, using SNNs to simulate the behavior of biological neurons. Despite these historical precedents, the full potential of ANNs for generating neural spike data has not yet been fully realized and warrants further exploration.

The main goal of this study is to generate neuronal spike data using one of the techniques described in Fig. 1 that can capture causal interactions between neurons. Beyond the naive methodology of using correlations between spike's data, we evaluated the similarities and differences between real and generated neural activities in terms of predicting future neural spikes.

The overview of the entire data processing flow in this study is summarized in Fig. 1.

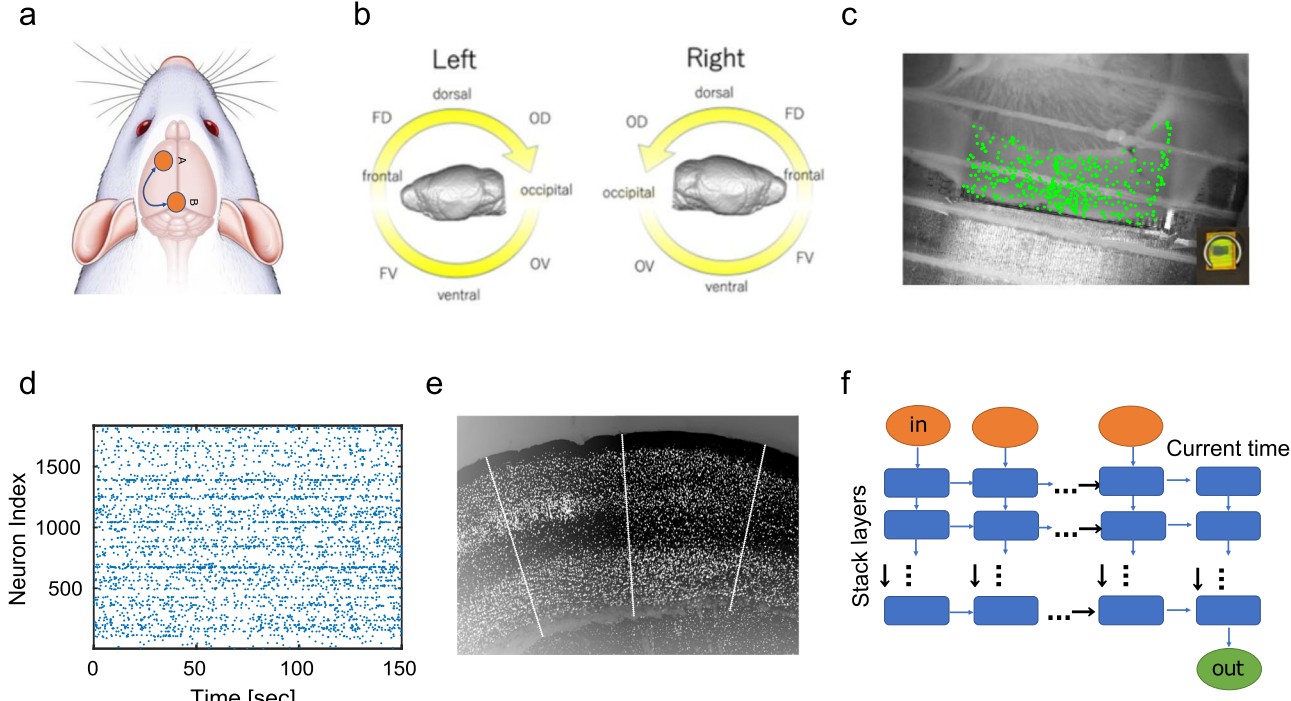

**Fig. 1 The workflow of this study. a** We prepared brain slices from individual regions, as illustrated here with orange circles. We used two of these regions for the training step and the generation step, respectively. **b** Prior to this, we classified the cortical regions into 16 groups. The abbreviated names of the 16 areas are defined by the following rules: Right and left hemispheres include 8 groups, respectively, and expressed as L or R at the beginnings of individual names. The name is followed by combinations of O, D, F, and V expressing abbreviations of Occipital, Dorsal, Frontal, and Ventral. (Refer to the supplemental material. 1 detailed locations of the slices used for the 16 area groups.) Pairs of regions, like the examples in **a**, were selected from those 16 groups. **c** We measured neuronal activity from hundreds of neurons in each region with a multi-electrode device. **d** An example of a spike train obtained from one of the measurements. The horizontal axis is time [sec] and the vertical axis is the index of neurons. The timing at which a certain neuron fires is indicated by a dot. This diagram is called a raster plot in neuroscience. **e** We used stained images to extract only neuron groups in cortical areas and then divided the neuron groups with lines orthogonal to the cortex so that only 128 cells were included in each dataset. **f** Such spike sequence, binary data, is input to the Multilayer LSTM model to predict one step ahead after learning from the past data. The horizontal axis is time [ms], and the input vector is a binary vector.

This study targeted the analysis of electrical activity of multiple neurons in the mouse cortex, mainly neocortex, measured with a multi-electrode array (MEA) (Fig. 1c, d). The neocortex consists of one to six layers, numbered from the surface in the direction of the depths. In each brain region, neurons were selected by sliding a section orthogonal to the cortical surface along the cortical surface so that all 1–6 layers were included, and a total of 128 cells were selected and collected into each regional data (Fig. 1e, Refer method about experiments in detail).

The neural activity recorded with an MEA is called spikes as mentioned before, which is represented as binary data, where elements with 1 describe firing timings.

In individual analysis, we prepared a pair of training and test spike data from two datasets. Training data is used to optimize the internal parameters of the multilayer LSTM model, enabling the best prediction within the training data. This is called the training process. After training is finished (Fig. 1f), we use the trained network to generate new spike data and compare it with the test data (Fig. 1a) (Refer to analysis method for more details). This is called the generating process. In the generating process, we generated one-step future neuronal spikes with hypothesizing that we know all neuronal spike sequences.

The test data is also sometimes referred to as target data because this data is the target of the generation process. The training and test data were respectively obtained from one of the 16 regions of the neocortex (caption of Fig. 1)[34].

The generated data were evaluated using the firing rate and the Synchronization score (refer to analysis methods). It is important to note that the ability to generate a highly predictive time series means that new future neural activity can be generated by extending the time from existing spike data. We thought that the above is important because it means that new time series data can be obtained without the need for new experiments, leading to fewer physiological experiments in the future.

However, to be honest, at the beginning of our research, we had a hypothesis that with this generation method, data from the same brain region would be able to generate each other effectively, but it might be challenging to achieve such effective generation between neural activities from different brain regions. Nevertheless, fortunately, as later results showed, there were instances where generation between different brain regions worked well. Based on these findings, particularly when evaluating time series generated by models trained with data from one brain region against test data obtained from another brain region, we hypothesized that the generation performance would likely depend on the relative "closeness" between the two brain regions. We also assumed that it would depend not only on the geographical distance but also on the strength of structural connectivity through white matter fibers. In our study, we conducted analyses in the final subsection results to explore this assumption.

## Results

**Training process**. The internal parameters of the Multilayer LSTM model were optimized to minimize the prediction error for ~17 minutes of training data.

As described in the Method section, based on the result of a preliminary study testing various parameterizations, we use a Multilayer LSTM network with three hidden layers, each with 128 LSTM units. We used a focal loss function to quantify the prediction error. The number of epochs for training was set to 350. This decision was based on the observation that even though the value of the loss function had converged before 25- epochs, the precision of the firing rate and the reproducibility of synchronous firing continued to improve up to 350 epochs [Fig. 2]. The Adam algorithm was used for optimization[35].

**Generation with the same region for source and target regions.** The neocortex was divided into 16 regional groups, with two datasets per regional group. Formally, the first 16 datasets are collectively named dataset 1 and the remaining datasets are named dataset 2. In the following sections, we observe the results of evaluating the data generated as a result of the training in various cases. Then, we present the average of the results obtained for these two datasets. The results presented in what follows are confirmed to be reproducible between the two datasets.

In this section, we first performed generation through the multilayer LSTM model by dividing the data given by the same group of brain regions in each dataset into the first half and the second half on the time axis, and preparing them as training data and test data, respectively [Fig. 3a, b]

Because the training and test data are different parts of the same time series, this case is relatively easy to generate and predict. Therefore, it was expected that it would perform close to the best prediction performance when the training and test data were cut out from the same time series.

Multilayer LSTM receives time series data of 128 cells in the past and outputs information on whether 128 neurons are active in the future (Fig. 1f). Multilayer LSTM was trained by swiping data in the first half of the time from time 0 to 17 min (Refer in more detail to the method sections about Multilayer LSTM). In the second half, the learning process is stopped, and the data is swiped from 17 min to 34 min to evaluate how well the rules learned in the first half can be used to predict future activity states. In other words, the similarity between the first half of the

data and the second half of the data is evaluated through the data generation performance.

To evaluate the quality of generated data, we computed the firing rates and synchronization scores for both the generated data and original test data, and analyzed how well their statistical properties were reproduced. The firing rate, defined for each neuron, refers to the number of spikes per unit time [spikes/sec]. The synchronization score, defined for each pair of neurons, represents the amount of deviation in firing frequency of a neuron during a certain time window after the other neuron fires. A positive value of this quantity indicates a co-occurrence relation between the neuron pair and a negative value indicates an inhibitory relation. See the section about Multilayer LSTM for precise definitions. A highly positive correlation of the firing rates or synchronization scores between the generated and real data indicates that the statistical properties of the real data are accurately reproduced in the generated data.

As a result, Fig. 3c, d show scatter plots between predicted and measured values for both the firing rate and the synchronization score, respectively. In these scatter plots, we observed a concentrated point on the diagonal for both the firing rate and the synchronization score, indicating that the generation was successful [Fig. 3c, d].

Figure 3g plots the correlation coefficients of the Firing rate for each brain region used in the measurement. In all regions, the correlation values exceed 0.9, indicating high predictive success, with the exception in RFV [refer to supplemental material. 1].

For further evaluation of the synchronization score, the first and third quadrants of the scatter plot were extracted and histograms were drawn in the direction of the rotation axis. Examples are shown in Fig. 3e, f. As seen in these results, the successful generation is reflected in the peaks in the histograms. From the histogram of synchronization score we calculate the sharpness in the first quadrant (refer Figs. 3e, 4e), which is calculated as the ratio of the area around the peak (width of $\pi/4$) to the area at other angles included in the first quadrant ($\Theta = 0\pi/2$). The sharpness in the third quadrant is also calculated in the same way in the third quadrant ($\theta = \pi 3\pi/2$) (refer Figs. 3f, 4f).

From these results, it was found that when the training data and the test data are obtained from the same region, both the firing rate and synchronization could be nicely reproduced (Fig. 3g, j).

In the next section, we will observe the case where the training data and the test data are obtained from different regions. The results given in this section provided us the approximate values of every prediction performance in the relatively easy problem of generating from training to test data cut out from the same time series (Fig. 3g, j). The performance would give us perspective on the highest value when generating across different data shown in the next subsection.

**Generation with all brain regions as source and target regions.** While training and generating evaluations were performed on data acquired from the same brain region in the previous section, in this section, we also analyze and evaluate the training and target data recorded respectively from two different brain regions groups included in the same one of two datasets [Fig. 4a, b]. The prediction performance was then shown as the average of dataset 1 and dataset 2 [Fig. 4g–k].

By comparing between different data, the similarity of their neural activity can be assessed by predictability by generating process rather than by cross-correlation. It should also be noted that in this in vitro experimental environment, the connections between the brain regions are broken, so the similarity between

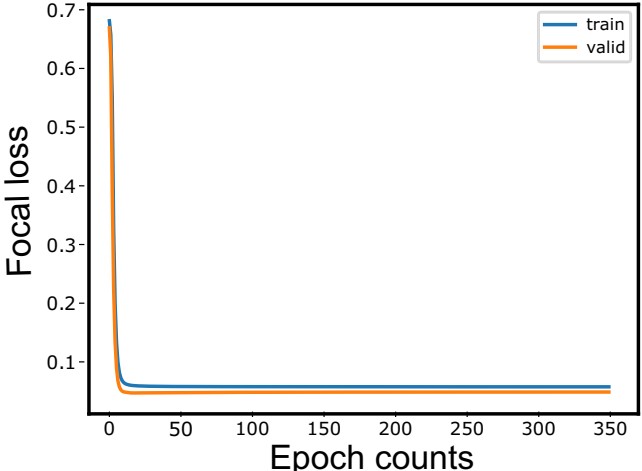

**Fig. 2 Learning procedure.** The figure depicts how the loss on the training data and the loss on the validation data decreases as the multilayer LSTM model is trained. A decrease in loss indicates that the training of the Multilayer LSTM has progressed. The loss, common to both data, decreases sharply at 25–100 epochs. However, the precision of the firing rate and the reproducibility of synchronous firing continued to improve up to 350 epochs.

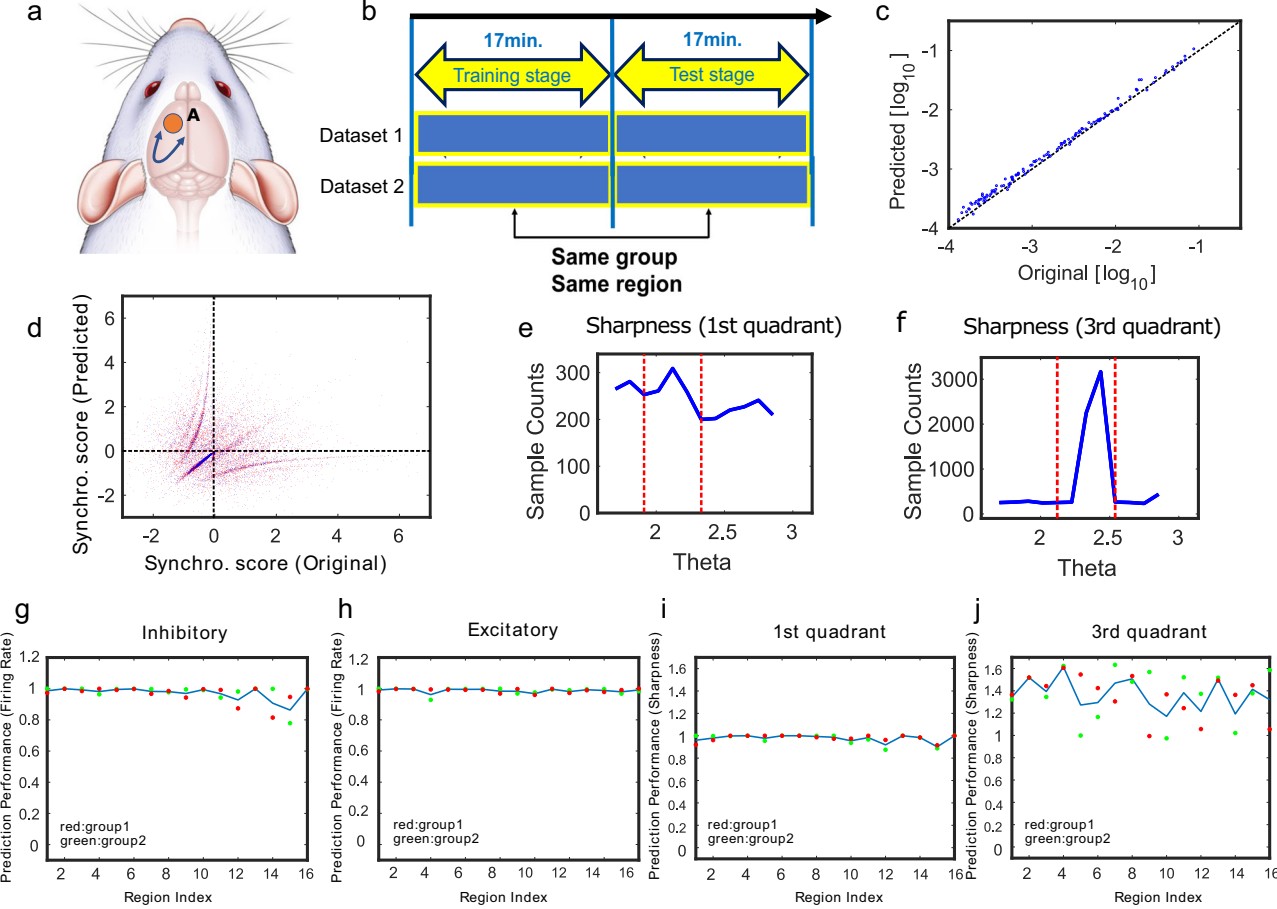

**Fig. 3 Results for generation when source and target are the same region. a** Both training data and prediction, generation, data are prepared from the same region in this evaluation. **b** In the case of example (**a**), both training data (first 17 minutes) and test data (second 17 minutes) of the same time series are obtained from area A. **c** This panel shows the result of predicting the firing rate in this case, where the x axis is the firing rate in the original data and the y axis is the firing rate in the data generated by training Multilayer LSTM. **d** This panel shows the result of predicting synchronization score. Again, the x axis is the synchronization score in the original data, and the y axis is the synchronization score in the generated data. This data was expressed in r-θ rotational coordinates, and a histogram of the number of data in 0-π/2 with respect to θ, or in the first quadrant, was drawn in **e**. Finally, **f** is the histogram of the number of data in π-3/2π with respect to θ, or the third quadrant. In particular, if the output is coming from inhibitory cells, it is distributed in the third quadrant. The sharpness of the peaks in these histograms (**e**, **f**) was evaluated by sharpness [Refer to the method section]. **g** Correlations between expected and true values of firing rates in the inhibitory neurons are plotted for every 16 regions. The two points for every group of regions correspond to the two datasets, and the line is the averaged value between the two datasets. The meaning of the points and lines is the same for **h**, **j**. **h** Correlations between expected and true values of firing rates in excitatory neurons are plotted in the same way as in **g**. **i** shows correlations between expected and true values of synchronization score in the first quadrant for every 16 regions. **j** shows correlations between expected and truth values of synchronization score in the third quadrant.

time series data from two brain regions is by no means produced by the interaction of the two brain regions.

Again, the quality of generation was evaluated using the firing rate (Fig. 4g–i) and the synchronization score (Fig. 4j, k) (Refer method section about Multilayer LSTM).

First, accuracy with respect to firing rate in generation was evaluated simply by cross-correlation between the firing rate in the original test data and the firing rate in the generated data (Fig. 4c). Color maps of the correlation values between the firing rates of all neurons (Fig. 4g), inhibitory cells only (Fig. 4h), and excitatory cells only (Fig. 4i) are plotted. Hierarchical clustering was performed to sort brain regions that show similar patterns in terms of prediction performance into close indices.

Second, when evaluating the accuracy with respect to the degree of synchrony in the generation, we used the scatter plot (Fig. 4d) between the synchronization score in the original test data and the synchronization score in the generated data.

Then, in the scatter plot, we evaluated the peakness of the angle-dependent distribution in the first (Fig. 4j) and third (Fig. 4k) quadrants of the data distribution as the sharpness (method section). In the synchronization results, the data were sorted by hierarchical clustering so that regions with similar characteristics are close to each other.

In all the results so far, the diagonal components are brighter than in other cases because the generation between the same region shows a high prediction performance. However, at the same time, the generation between different regions also sometimes showed high prediction performance at the same level as the generation from the same region.

In the two sections ahead, we will further analyze how such brain region pairs of non-diagonal cases, showed similar prediction performances with the diagonal cases are related to each other, based on the relative spatial distances (angles) and/or structural connectivity between brain regions.

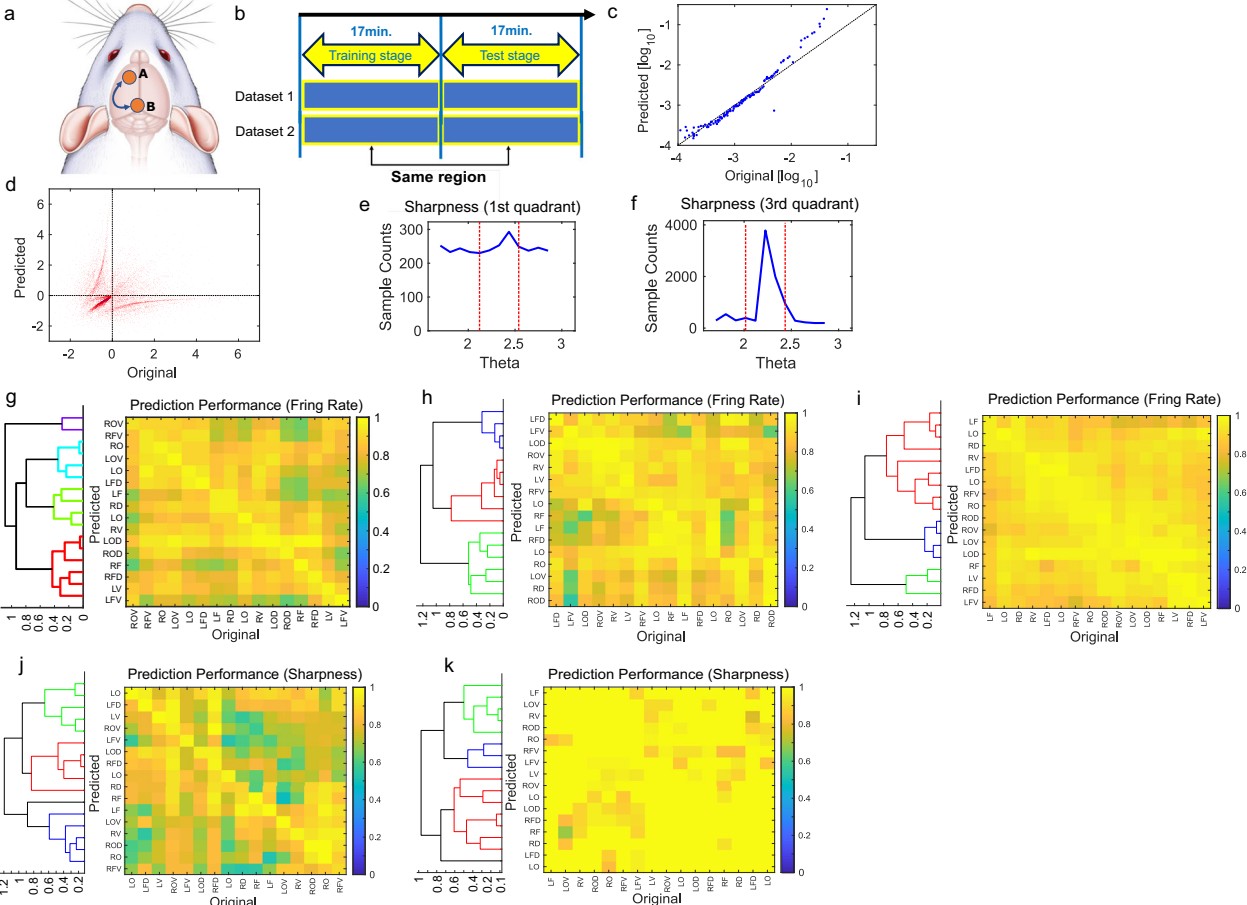

**Fig. 4 Results for generation across all pairs of brain regions. a** In this evaluation, training data and prediction, generation, data are also prepared from different regions. **b** In the case of **a**, training data (first 17 minutes) is obtained from region A and test data (second 17 minutes) from region B. **c** This panel shows the result of predicting the firing rate in a particular case, where the x axis shows the firing rate in the original test data and the y axis shows the firing rate in the generated test data by training Multilayer LSTM. **d** This panel shows the result of predicting the synchronization score in a particular case. Again, the x axis is the synchronization score in the original test data, and the y axis is the Synchronization score in the generated test data. This two-dimensional distribution was expressed in terms of r-θ rotational coordinates, and panel. **e** depicted the density distribution of the number of data in 0-π/2 with respect to θ, i.e., in the first quadrant. Finally, panel **f** is the density distribution of the number of data in π-3/2π with respect to θ, or the third quadrant. In particular, we have confirmed that the distribution is restricted to the third quadrant when the output is from inhibitory cells. The sharpness of the peaks in these histograms (**e**, **f**) was evaluated by sharpness. **g** is the correlation between generated and truth values in firing rates for all cells, **h** for inhibitory cells, and **i** for excitatory cells. x axis is the region index of the original data and y axis is the region index of the predicted data. The correlations of firing rates in all those pairs are plotted as color maps. In addition, these color maps are sorted based on hierarchical clustering. **j**, **k** are the color maps at sharpness in the first and third quadrants, respectively. The point that the sorting is based on hierarchical clustering is the same as in the case of the color map of firing rates.

**Relationship with connection strength and relative angle.**
Finally, in order to study how anatomical "closeness" relates with the unevenness of performance in intergeneration between different brain regions in the results obtained in the previous section and the one before that, we evaluated the results in comparisons to the relative angles between brain regions and the strength of structural connections.

As shown in Fig. 5a, relative angles were calculated based on the relative angles from the regions of interest selected from 16 regional groups. First, the relative angle in the group of interest was set to zero. Then, within the ipsilateral cortex of the group of interest, the relative angle was incremented by +1 with every one-angle difference. However, the completely opposite angles were set to +2 since they are adjacent to each other on the same slice plane. (Fig. 5a).

We utilized tracer data published by Allen institute with the Mouse Reference atlas for the structural connections[36–38]. In this study, we calculated the connection strength between two square recording regions in four steps. : First, we enumerated the cortical areas on the atlas that belonged to the two square recording regions where electrical activity was measured. Second, we calculated the strength of the structural connections between all pairs of the cortical areas belonging to each square region. Third, the connection strength was normalized by the percentage of area within the region located at either end of the connection. Finally, the normalized connection strength was averaged for all combinations of regions and calculated as the connection strength between electrode recording regions (Fig. 5b). See Method section about analyses for the detailed formula.

Then, we evaluated the relationship between either the relative angle or the connection strength between the electrode recording regions and either the accuracy of predicting the firing rate or the sharpness and accuracy of predicting the synchronization score between the region pairs. This evaluation was performed with the left and right hemispheres separately (Fig. 5c–v).

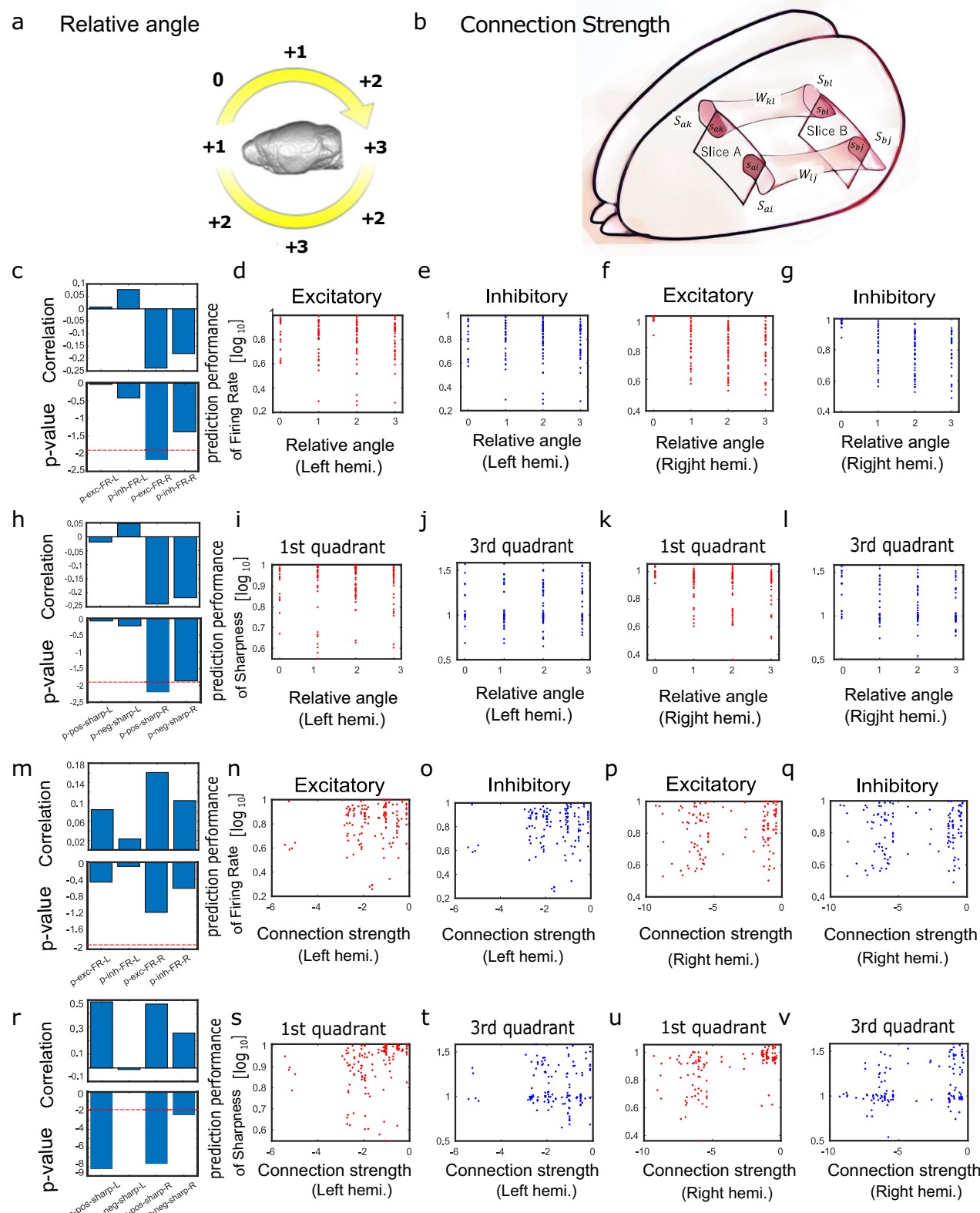

**The statistical test here is a Bonferroni correction for a sample size of 4 in the panel**. First, Fig. 5c–l lists the results for relative angle. For a difference of 1 relative angle from 0, the predicted firing rate showed a significant difference (Fig. 5c–g) ($p = 0.006$, $p < 0.01$, Bonferroni correction). However, sharpness, which is the prediction performance of connection strength, in relation to relative angle showed no significant trend.

Second, Fig. 5m–v lists the results for the connection strength. No significant trend was observed in the prediction of firing rate for connection strength in any condition (Fig. 5m–q). However,

**Fig. 5 Comparison between generation performance and spatial relative position or structural connections. a** shows the definition of a score calculated based on the relative angle from a certain region of interest (in this case, the Left Frontal Dorsal region). The score at the region of interest was set to zero, and the score was added by one for every shift of one angle from the region of interest. However, the score for the region completely opposite to the region of interest was reduced to +2 because that region is located at the same slice surface as the region of interest and adjacent to the region of interest. **b** illustrates the definition of the values required to calculate the strength of the structural connections between the square recording regions where electrodes were placed. We downloaded the original connection strengths and atlas data from open data shared by Allen institute (https://connectivity.brain-map.org), and reconstructed the connection matrices. In this illustration, the two squares represent the areas (e.g., Slice A, Slice B) where electrical measurements were made. In addition, two examples of connections between them are expressed as pipes. The connection strength between the atlas areas included in the square region of the electrical recording was calculated for each connection as a normalized value in terms of the percentage of the area, expressed as a dark red area in Fig. 5b, of the intersection of the region of the electrical recording with the area of the atlas at both ends of the connection. The normalized value as a percentage of the area, expressed as a dark red area in Fig. 5b, was calculated for individual connections (Refer to the method section for details). In the square region of the electrical recording, the normalized value, expressed as a dark red area, was calculated for each connection as a percentage of the area of the intersection of the region of the electrical recording with the area of the atlas at both ends of the connection). The normalized value as a percentage of the area was calculated for individual connections (Refer to the method section for details). Then, the connection strength between the regions of electrical recording was calculated as an averaged quantity of connection strength in all connected pairs of atlas areas at both ends. **c-l** plot the prediction results for the relative angles between the measurement regions. Among them, **d-g** plot scatter plots with the predicted firing rates for the relative angles between the regions, and **c** summarizes the correlation values and *p* values for the points between Angle = 0 and Angle = 1 in **d-g** as a four-bar graph corresponding to the order from **d-g**. Among **d-g**, **d** represents results to excitatory neurons in the left hemisphere, **e** to inhibitory neurons in the left hemisphere, **f** to excitatory neurons in the right hemisphere, and **g** to inhibitory neurons in the right hemisphere. **i** through **l** are plotted as scatter plots with sharpness, prediction performance of Synchronization score, as the vertical axis relative to the relative angles between the measurement regions. The difference in the meaning of the *x* axis in the **i-l** means results to the first quadrant of the left hemisphere, **j** to the third quadrant of the left hemisphere, **k** to the first quadrant of the right hemisphere, and **l** to the third quadrant of the right hemisphere. **h** The correlation values and *p* values for the points at Angle = 0 and Angle = 1 in **i-l** are summarized as bar graphs. **n-q** and **s-v** are the same as the **d-g** and **i-l**, except that the horizontal axis is the connection strength. Then, the correlation values and *p* values at **n-q** are summarized in **m**, and the correlation values and *p* values at **s-v** are summarized in **r**. The significance level is set at about *p* = 0.01 and the dotted lines are overlaid, and it can be read that the *p* values corresponding to **s** and **v** are much lower than that level.

there was a significant positive correlation between connection strength and predicted sharpness (Fig. 5s, u) in the first quadrant (Left hemi.: $p = 5.5 \, 10^{-9}$, Right hemi.: $p = 2.0 \, 10^{-8}$, $p < 0.01$, Bonferroni correction), which was common in the left and right hemisphere.

In general, the results indicate that the prediction performance of the index of firing activity in the multilayer LSTM is related to the relative angles between measurement sites, and the prediction performance of synchronization is related to both of relative angles and the strength of the structural connections. We will address more in-depth discussions of this relationship in the discussion section.

## Discussion

In this study, we developed an approach, novel to the best of our knowledge, to evaluate the homology between two regions through the generation and evaluation of synthetic neural spike data using a Multilayer LSTM network.

Specifically, spiking data for a group of over 100 neurons were measured from slices taken from 16 cortical regions of the mouse, and from those 16 regions, $16 \times 16 = 196$ different pairs of training and test data were prepared.

When interpreting the given results, it is important to keep in mind that in slices cut from one region, connections to other regions are physically severed. It was well expected that our applied spike generation technique between different regions would not work at all because the recording regions are disconnected from each other.

However, surprisingly, the results showed that there are hidden rules in the spike data that allow the Deep Neural Networks used in this study to generate complex spike sequences to the point of reproducing them with nontrivial accuracy. It is extremely difficult for the human eye to decipher the rules utilized in their generation.

It should also be emphasized that even if one creates a detailed computational model of neural systems, it is actually very difficult to prepare a computational model that generates spikes that

somehow reproduce the synchrony among the many pairs of neurons in the system[39–41]. The findings of this study can be summarized in the following three categories:

First, the case of learning and generation among time series of different time periods in the same data showed clearly significant prediction performance, not only in terms of firing rate, but also in terms of the degree of synchrony.

Second, even in the predicted performance among the regions measured from different regions, surprisingly, there were some combinations that came close to the performance for the same data.

This indicates that the characteristics of electrical activity within cortical local circuits have enough commonality or universality to generate each other even if the regions are different. There is no precedent for showing this commonality through the mutual generation of activity.

Third, we compared the prediction performance of firing rate and synchronization with the relative angle between the measured regions and the strength of the structural connections. The results showed that there was a significant difference in the prediction performance of firing rates between generations made between the same region and those made between regions that were one relative angle adjacent to each other. Moreover, and more surprisingly, although no significant correlation was observed between structural connection strength and prediction performance of firing rate, significant correlations between structural connection strengths and sharpness, which is prediction performance of synchronization score, were observed in both left and right hemispheres or in one hemisphere in the region of the first quadrant.

We need to notice that it is also important to keep a dispassionate attitude in considering observed high prediction performance. For example, in the relation between the prediction performance of the firing rate and the relative angle, there was a significant difference in the prediction performance of the firing rate between the case of generations made between the same regions and the case of generations made between one adjacent

region. Remember, however, that there is a possibility that the advantage of being cut from the same data (beyond being in the same region) worked in the prediction between the same regions. Therefore, further verification is necessary to check if it is not an artifact. Nevertheless, the trend in the first quadrant that synchronization score increased with each increase in structural connectivity is a nontrivial trend that cannot be explained by such reasoning.

Although this is not a result obtained in this study, it should be noted that there have been numerous previous studies on structural connectivity. First, the hierarchy of information processing, which is described in terms of structural connectivity patterns between cortical areas, has been logically defined based on differences in laminar patterns along the cortical depth direction. This is the definition conventionally employed in the analysis of wide-ranging structural connectivity patterns, mainly for the visual system[42].

Some researchers have attempted to understand the hierarchy of information processing by integrating this pattern with the auditory and somatomotor systems in informatic ways[43]. The understanding of the hierarchical structure of information processing was then extended to an understanding of hierarchy in the sense of going from peripheral areas closely connected to the peripheral nervous system of information processing to central core areas corresponding to the association cortex[44].

We can also confirm that the hierarchy of information processing reflected in connection structure patterns is related to cell density[45]. Furthermore, as we obtained better systematic data on connection strength, it became clear that there is a clear empirical relationship between connection strength and spatial distance[46].

Various studies have reported that the pattern of structural connections is similar to the pattern of functional connections defined on the basis of synchronization of activity between joined brain regions[47–49]. The investigation of how such characteristics are related to the activity generation between disconnected brain regions, as observed in this study, will be a future task and will be discussed in the next subsection.

Even with the important background knowledge described in the above subsections, it is quite surprising that we were able to achieve high performance in generating activity between different brain regions, and also that we observed a significant positive correlation between structural connection strength and sharpness for the first quadrant (for example, Fig. 5s). Our experiments involved measuring neural activity from brain regions after sectioning them as slices. This means that the connections between those brain regions are severed, and as a result, these regions do not have shared input. Therefore, the factors from outside these two regions that would normally preserve the similarity in neural activity between them are absent.

Structural connectivity patterns in mice have been measured and analyzed on a large scale with increased resolution in a way that also integrates with genomics or transcriptomics studies[36,50,51]. This study also aided the structural wiring pattern information obtained in those studies[38].

Such genes and transcription factors are internalized characteristics of each brain region. In other words, these studies indicate that brain regions connected by structural wiring possess similarities in terms of their internal activity generation characteristics. In our research findings, the reason for successful generation between disconnected brain regions is understood to leverage the similarity of activity generation characteristics that each brain region inherently possesses. In essence, connecting the dots, there is a possibility that the similarity of genes and transcription factors is related to the ease of mutual generation of neural activity in each brain region, which we have discovered.

Therefore, it is a future challenge to investigate the relationship between these genomics and transcriptomics and the internal characteristics of activities within individual brain regions. To understand the connection between genes, transcription factors, and neural activity characteristics, it is crucial to continuously grasp the multi-layered hierarchy of how proteins produced from genes govern the chemical properties of neurons, such as their structural and chemical channel characteristics, leading to the emergence of electrical properties specific to each cell group.

One of the other issues is the improvement of the generation method using multilayer LSTMs and the evaluation method. When generation does not work well, the disappearance of the diagonal component in the scatter plot of the relationship between the predicted and correct answers generally occurs. Although this report does not go into detail, several characteristic patterns were observed following the disappearance of the diagonal component in the scatter plots. By classifying these characteristic patterns and exploring their causes individually, guidelines for their generation and evaluation at high performance will be more mature than now.

In this study, we chose the region that brings together the cell groups as the square recording region in the electrical measurements. In other words, the brain regions of the atlas provided by the Allen institute were grouped together within the square measurement area, and the analysis was performed to compare them with the structural connections. Another future task is to analyze cell groups separately according to the atlas brain area segmentation provided by the Allen institute. This will allow for a more stable combination of cell groups, including those within a single region, which will positively affect the generation and prediction results. -This could have a positive effect on prediction performance.

Furthermore, in this study, we selected the same 128 cells within the recording area for each of the 16 regions, which resulted in varying cell densities and spatial sizes proportional to each other across regions. However, some form of normalization is necessary for the analysis, and we believe that the chosen approach in this study is a natural method among the possible choices. In future research, we aim to conduct analyses where the number of cells varies across regions while keeping the spatial size fixed. Then, new ideas will be also necessary to deal with the fact that the number of cells in each region differs due to the difference in size of each region.

This study collectively analyzed populations of neurons existing in the square region used for electrical measurements. However, there may be cases where histologically distinct brain areas coexist within these square recording regions. Therefore, one of the future tasks is to analyze the neuron groups according to segmentations of brain atlases such as the Allen brain atlas. This will allow us to include only neuron groups within a single region, and to make the groups of neurons belonging to each individual group more uniform. As a result, we expect that this will have a positive influence on the performance of the generation and prediction. However, new ideas are necessary because the number of cells within a region should be different due to differences in sizes of these regions.

Utilizing more recent neural network architectures than the multilayer LSTM model also has the potential to improve the performance of data generation, even though new models do not necessarily improve performance due to the size and characteristics of the data. For example, the Transformer model[52], which uses the self-attention mechanism to deal with correlation in sequential data, can potentially learn even longer temporal correlations than the LSTM model.

This study showed that the activity of multiple neuron groups in the cortex can sometimes be generated reciprocally, even

between different regions. We also showed that the non-uniformity of the reciprocal generation can be explained, to some extent, by the relative positional relationships and structural wiring.

The methods developed in this study have very important significance for the fundamentals of animal experimentation. In neurophysiological experiments, synchronization still plays an important role in quantifying neural representations of neural interactions and cognitive functions[53–56]. If such spiking data can be generated from artificial models with high accuracy, there is no need for redundant experiments. As a result, they can contribute to the 3 R principle, Replacement, Reduction and Refinement, regarding animal experiments[57]. Such techniques will become even more important for rare data, where large amounts of data are difficult to obtain[58,59]. Therefore, it is likely that research schemes to quantify the similarity of different datasets measured under different conditions in terms of their inter-generational capabilities will expand in the future.

In such a method, the combination of the original data and the target data to be generated is very broad. In the future, as the physiological interpretation is deepened and the prediction performance is improved, it will become the basis of a method to "measure" physiological data through mathematical models instead of experiments, and is expected to contribute to the comparison of experimental data among animal species and the performance evaluation of model animals as well as the 3Rs. This method is expected to contribute not only to the 3Rs but also to the comparison of experimental data among animal species and the evaluation of model animal performance.

In this study, we demonstrated for the first time that the activity of groups of multiple neurons in the cerebral cortex can often be mutually generated, even between different brain regions. We also suggested that the non-uniformity of performance in mutual generation can be somehow explained by relative spatial positions and structural connections.

It should be emphasized that such a generation method allows a very wide range of choices in what data to use for the combination of original data and target data to be generated. In other words, this method allows us to evaluate similarities between a wide range of neural activities.

## Methods
### Experiments
*Data acquisition of neuronal activities.* We used neuronal spike data recorded and studied in detail in our past study. Here, we briefly explain the experimental procedure utilized in the past study[34,60,61]. The whole experimental processes are also now open in a video journal[62].

We used female C57BL/6 J mice ($n = 32 = 16 \times 2$, aged 3–5 weeks). This study grouped the cortex (mainly the neocortex) into 16 groups, and prepared two sets of data for each of these groups (Fig. 1b, supplemental material. 1). All animal procedures were conducted in accordance with the guidelines of animal experiments of Kyoto University (KU), and have been approved by the KU Animal Committee.

This study recorded neuronal spikes from cortical slices with a MEA system (Maxwell Biosystem, MaxOne) with refluxing an artificial cerebrospinal fluid (ACSF) solution that was saturated with 95% O2/5% CO2[60,62].

Prior to slice preparation, mice are thoroughly anesthetized (1%-1.5% isoflurane), cervical vertebrae was dislocated, and brains are removed. We immersed the removed brain in a cutting solution, an ice-cold solution used to prevent brain deterioration, and bubbled it with oxygen continuously. The brain was cut with a vibratome (NLS-MT, DOSAKA EM CO., LTD), scanned, and

sliced by a slicer in the target region of the brain. 300-μm slices were made by changing the height of the cutter. The slices were immersed in an ACSF solution, saturated with 95% O2/5% CO2, for one hour before electrical measurements were taken.

The recording area of the MEA used was $1 \times 2$ mm$^2$, and 26,000 electrodes were uniformly arranged at 15-μm intervals (on the order of cell spacing distance). This high-density electrode arrangement enables accurate determination of neuron location. For the main measurement, we used on the 1020 electrodes, selected, as receiving strong input from the neurons, in the 20-minute pre-scan. The pre-scan refers to a procedure conducted before the main recording, where we restrict the placement of electrodes only to regions where activity is likely to be observed. This procedure is implemented in the application of the device we are using to effectively perform the main measurements. The number of electrodes, 1020, is sufficient for placing electrodes around the active cells in the $1 \times 2$ mm$^2$ recording area within the slice.

We performed spike sorting (Spyking Circus software) from the time series obtained in the main measurement and converted to time series binary data of the activity of the cell population. Refer to the following references about the details of the experimental procedure[34,60,62,61]. In these papers, we also use stained image data to extract 128 cells included in the region enclosed by two lines in the depth direction of the cortex, covering all layers 1–6 of the cortex, and the surface and the deeper side. Each cell's time series is then fed as input to one node of the LSTM model. This aspect is described in detail, particularly in ref. [61].

*MRI acquisition.* This study measured 3D T-weighted (T2W) images of the whole brain in each mouse. For this purpose, a 7 T, 210 mm horizontal bore, preclinical scanner (BioSpec 70/20 USR, Bruker BioSpin MRIGmbH, Ettlingen, Germany) MR system equipped with a 440 mT/m, 100 μs ramp time gradient system was used for relaxation enhancement (RARE) sequence was used; for RF excitation and signal reception, an orthogonal volume resonator (35 mm i.d., T9988; Bruker BioSpin) was used.

We also used a protocol called TurboRARE-3D (Bruker BioSpin), and the specific acquisition parameters are as follows. : Repetition time (TR) 2000 ms; Echo time (TE) 9 ms; Effective TE 45 ms; RARE factor 16; Acquisition matrix size $196 \times 144 \times 144$; Field of view (FOV) $19.6 \times 14.4 \times 14.4$ mm; Acquisition bandwidth 75 kHz, axial (coronal direction in scanner setting): bandwidth 2.6 ms-Gaussian $\pi/2$ pulse for fat suppression and spoiler gradient with 1051 Hz bandwidth, 2 dummy scans, averaging number 3, acquisition time 2 h 42 m, excitation pulse 2.59 ms, re-convergence pulse 1.94 ms, pulse shape: $\pi/2$ pulse, bandwidth 1051 Hz, fat suppression averaging number was 3. Pulse shape was sinc3, bandwidth was 2400 Hz.

The software for the measurements was ParaVision 5.1. The cortical surface images were extracted from the measured MRI images using FSL. For details, see Ide et al. 2020.

*Slice preparation and electrophysiological recording.* In this study, we recorded neuronal spikes from neocortical slices using the MEA system. For this purpose, mice were first sufficiently anesthetized with 1%-1.5% isoflurane, then transferred it to a petri dish (100 mm × 20 mm) containing ice-cold cutting solution with air flow containing 95% O and 5% CO. The extracted brain was cut into two blocks and then sliced into 2–5 slices (300-μm thick) in a diagonal angle to the cortical surface using a vibratome (Neo Linear Slicer NLS-MT; DOSAKA EM CO., LTD).

We selected cutting speed, frequency, and swing width as 12.7 mm/min, 87-88 Hz, and 0.8-1.0 min, respectively, and

slowed the speed down if the brain needed to be cut carefully for a temporary period.

All slices analyzed in this study were taken oblique to the cortical surface in any region. The angle cutting slices was carefully chosen, and the coordinates of the sections, including anterior-posterior coordinates and hemispheres, were recorded in a format when the brain sections were cut, and the slices were also reconfirmed and accurately recorded by embedding them within the MRI space acquired during the MR measurements described above [Ide et al., 2019].

The slices were incubated for 1 hour in a beaker filled with pre-warmed ACSF (~34 °C), then we selected one slice per animal and moved with a thick plastic pipette onto the MEA array and positioned the slice with a soft brush to record properly from a specific brain region including the cortex. The MEA array is rectangular in shape (2.0 mm × 4.0 mm), with 26,000 electrodes uniformly distributed and the distance between adjacent electrodes was 15 μm (Maxwell Biosystem, MaxOne; https://www.mxwbio.com/).

Prior to the main electrical recording, a so-called pre-scan of 30 [sec] was performed by recordings from adjacent 1020 sensors combinationally covering the all sensors, and up to 1020 sensor sensors that responded more strongly than 0.03 mV and more frequently than 0.1 Hz were selected.

The main electrical recording of spontaneous neural activity from the selected sensors was then performed for ~2.5 hours. In our experimental setting, the firing rate did not decay in this 2.5 hours. This long recording is an important factor in achieving high performances of connectivity estimation.

During this pre-scan and main measurements, the slices were still perfused at 1 mL/min with ACSF that was saturated with 95% O/5% CO while controlling the temperature of the perfusate around 34 °C.

*Brain surface scan*. We recorded the brain surface in three different 3D scans: the whole brain immediately after extraction, the brain block cut into two blocks, and the brain block remaining after slicing into slices. For each object, the brain itself is measured at least twice, flipping up and down as needed. In addition, between one measurement from the same object, 16 images are scanned by rotating the object. For its 3D scanning, a scanning system based on 3D structural optical technology (SCAN in a BOX; Open Technologies) was used. Before scanning, it is extremely important to lightly wipe the brain surface with a microfiber cloth to prevent diffuse reflections[62]. The scanned images were processed using the 3D scanning and processing software IDEA (including SCAN in a BOX).

First, the automatic alignment option of that IDEA was used to correct small disagreements between its 16 images, and integrated them. The merged image was obtained for the number of times it was recorded from the same object.

Next, the merged images were superimposed and merged using the manual alignment option. The optimization algorithm was an iterative closest point (ICP) algorithm without nonlinear deformation, which is often applied to rigid 3D objects[63]. After scanning, all scan planes were verified to be nicely overlapped.

Third, a high-resolution meshed object was created from the merged objects using the mesh generation option, saved in stl-binary format, and superimposed on the mesh image of the brain surface obtained from the individual MRI images using IDEA.

By the normalization process of FSL, the cross sections of the superimposed brain slices are grouped together in the same normalized brain space.

*Arrangements of data formats to input into the analytical model.* The spike data can be represented as a binary matrix $X_{ti}$ where t

and i are time and neuron indices: if neuron i is firing at time t, then $X_{ti} = 1$, and otherwise $X_{ti} = 0$. We sorted the neurons so that inhibitory neurons have earlier indices than excitatory neurons. Within each set of neurons, neurons are sorted in the order of layers (from 6 to 1). Each segment of spike data was split into training and test data. After removing the first 30 minutes, two 17-minute-long segments, earlier segment (30–47 min) and later segment (48–65 min), were cut out. The reason for excluding the first 30 minutes is that during the observation of raw data, it was confirmed that there are data points where the firing rate is not stable within the initial 30 minutes of the experiment. In the same region cases, we used the earlier segment as training data and the later segment as test data. In contrast, in the different region cases, one of the earlier segments from the two regions was used as training and the other one as test data.

### Analysis

*How to connect different data?.* In this study, we use a Recurrent Neural Network (RNN) with long short-term memory (LSTM) units for generating spike data. Hochreiter, Schmidhuber[64]. An RNN takes time series data as input and outputs time series data of the same length, and it iteratively processes a time-sliced (vector) data at a time instant. Thanks to its design, the information accumulated from the past data is to be utilized along with the current input.

Simple RNNs have difficulty for dealing with long-range dependence and have limited ability to learn the influence of data from the distant past. The LSTM units are proposed to alleviate this problem by introducing an architecture to deal with long-term memory. An LSTM unit contains three gates: "input", "output" and "forgetting". These gates determine the degree to which information is allowed to pass through depending on the conditions. The LSTM unit can store information more efficiently by gradually changing the long-term memory while maintaining the RNN structure itself.

Here, let us describe the history of LSTM. In 1997, Hochreiter and Schmidhuber et al. proposed LSTMs with cells and input and output gates, and in 1999 Gers et al. introduced an oblivious gate in the LSTM structure. In 1999, Gers et al. introduced an oblivion gate in the LSTM structure, which allows the LSTM itself to reset its own state. In 2000, Cummins et al. added peephole connections to allow cell-to-gate coupling; in 2014, Cho et al. proposed a gated regression unit, and a subsequent speech recognition using LSTM showed a 95.1% recognition accuracy. The LSTM network has been applied to speech recognition[65], language modeling[66], and many other tasks. The model is still widely used today.

We utilize a multilayer LSTM network, which can learn even longer-range dependence on the data than a single-layer LSTM network can (Fig. 6). The following hyperparameters of the LSTM network, which are different from the parameters to be trained from data, were used in this study.

First, the number of layers in the LSTM network is five, including the input and output layers. The number of LSTM units, which is the dimension of data in the intermediate layers, was 128 (same for all intermediate layers). The dimension of the input and output vectors, which are the number of neurons, was also 128.

The input of our LSTM network at time t is the vector $X_t$ = ($X_{ti}$), and the output at time t is the vector of probabilities $\pi_{(t+1)i}$ that neurons i will be firing at (t + 1) (each element $\pi_{(t+1)i}$ takes a real value between 0 and 1).

The network is trained with a loss function defined in the next section.

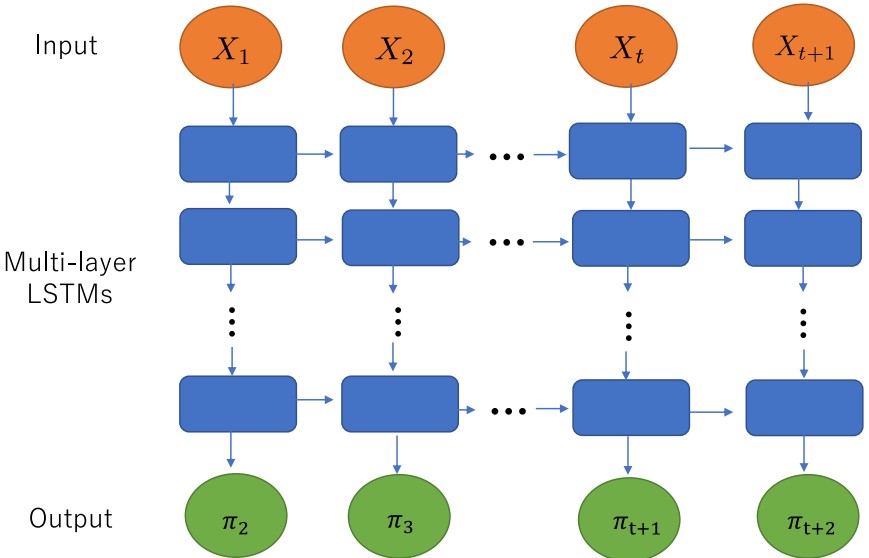

**Fig. 6 Muti-layer LSTM model.** This study utilizes multilayer LSTMs (multilayer LSTMs), which are networks of LSTMs layered on top of each other to allow for even longer-term learning than single-layer LSTMs. The input is a prediction of the probability of firing, and the output is the result of the target prediction. Whether or not a target has fired is determined by whether or not the measurement results exceed a threshold value.

After training, to generate spike data, we use the test data as input and apply thresholding to the output $\pi_{ti}$: neuron i is regarded to be firing at time t if $\pi_{ti} > h$, where h is a threshold value. The whole code we used is shared at https://github.com/ShimonoMLab/GenerativeNeurosci_ML-LSTM/.

*Loss function.* The binary cross entropy loss function is often used for training a neural network for data generation.

It is formulated as

$$L_{CE} = -\Sigma_{t,i}\left[X_{ti}\ln\pi_{ti} + \left(1 - X_{ti}\right)\ln\left(1 - \pi_{ti}\right)\right] \quad (1)$$

where $\pi_{ti}$ denotes the network's output and $X_{ti}$ denotes the training data.

In the spike data, most elements in $X_{ti}$ are 0 since the firing rates are small. In such a situation, the binary cross entropy loss often makes the network learn to output very small firing probabilities, which hinder the appropriate generation of spike data. To deal with such imbalances between states, a refined loss function called focal loss has been proposed[67].

The loss function has an additional parameter $\gamma$ and is given as

$$L_F = -\Sigma_{t,i}\left[X_{ti}\left(1 - \pi_{ti}\right)^{\gamma}\ln\pi_{ti} + \left(1 - X_{ti}\right)\pi_{ti}^{\gamma}\ln\left(1 - \pi_{ti}\right)\right] \quad (2)$$

Here, the parameter $\gamma > 0$ induces the two terms to balance since the non-firing probabilities $1 - \pi_{ti}$ are much larger than the firing probabilities $\pi_{ti}$.

We use the value $\gamma = 2$ as suggested in the original study[67].

The number of epochs for training was set to 350. We chose this number because, although the loss value had converged in about 150 epochs, the prediction performance of the firing rate and the reproduction performance of synchronous firing improved as the training further proceeded. The Adam optimizer[35] was used for training. The batch size, which is the number of data segments used in one update of the training process, was set to 64.

*Evaluation of similarity between generated and real data.* We analyzed the probability of synchronous firing between neurons to evaluate generated data with respect to the reproducibility of the property that the timings of neuron firing are synchronized between two cells with a specific delay. To formulate an

evaluation metric, suppose a situation where after neuron i fires neuron j fires with an acceptable delay D.

If the firing of neuron i has a positive effect on the firing of neuron j, then we expect a larger firing probability of neuron j within some acceptable delay D than its mean firing probability (i.e. firing rate).

Let us express the conditional probability that neuron j fires at least once within delay D after neuron i fires as $q(j|i; D)$. If neuron i and neuron j are not synchronous (i.e. they are independent), the conditional probability is given as

$$q(j|i; D) = 1 - \left(1 - p_j\right)^D = \bar{q}(j; D) \quad (3)$$

where $p_j$ expresses the firing rate of neuron j. We used the following indicator Z to evaluate how much the actual conditional probability $q(j, |, i; D)$ is biased from the null hypothesis $\bar{q}(j; D)$:

$$Z(j|i; D) = \frac{C(j|i; D) - N_i\bar{q}(j; D)}{N_i\bar{q}(j; D)\left[1 - \bar{q}(jD)\right]} \quad (4)$$

where $N_i$ is the number of firings of neuron i, $C(j|i; D) = N_i(j|i; D)$ is the number of times that neuron j fires at least once within delay D after neuron i fires, and the denominator represents the standard deviation in the null hypothesis.

We call this quantity synchronization score. We obtained a scatter plot of the synchronization score plotted with the original test data on the horizontal axis and the generated data on the vertical axis (refer to Figs. 3d, 4d), and a histogram with the rotation angle θ from 0 degrees as the main axis.

From the histogram of synchronization score we calculate the sharpness in the first quadrant (refer to Figs. 3e, 4e), which is calculated as the ratio of the area around the peak (width of $\pi/4$) to the area at other angles included in the first quadrant ($\Theta = 0 \sim \pi/2$). The sharpness in the third quadrant is also calculated in the same way in the third quadrant ($\theta = \pi \sim 3\pi/2$) (refer to Figs. 3f, 4f).

Just before detecting those peaks, we performed a linear regression on the histogram, and only the trend of the slope of the line was removed. The acceptable delay D was set to 1 ms. The reason is that the sharpness for other values of D had a negative

effect on observing the relationship between the generated and real data, blurring the diagonal components of the scatter plot than when D = 1.

*Definition of relative angles between recording regions.* The angles between regions were classified into eight groups in 45-degree increments in the direction of rotation with the line connecting the left and right ears as the axis. In other words, the left and right hemispheres were taken into account and classified into $8 \times 2 = 16$ groups (Fig. 5a). Note that in this study, the angle between two groups is also called the relative angle between two groups. 16 groups are named as explained in the caption of Fig. 1, with two data belonging to each region (Refer to the supplemental material 1).

*Definition of connection strength.* In this study, we superimposed the cutout position of each slice on the Allen reference atlas (ver.3, https://mouse.brain-map.org/static/atlas) and extracted the brain region in the Allen atlas that each slice includes.

Now, as shown in Fig. 5b, we consider two slices and call them slice A and slice B.

Then, focusing on brain regions $a_1, a_2, \ldots, a_n$ included in slice A (divided by the atlas) and brain regions $b_1, b_2, \ldots, b_m$ included in slice B, We obtained the connection strength $W_{ij}$ between $a_i$ ($i = 1 \sim n$) and $b_j$ ($j = 1 \sim m$) for all pairs.

Next, calculated the ratio $R_{ai} = s_{ai}/S_{ai}$ ($i = 1 \sim n$) of the area $S_{ai}$ of the entire brain region on the slice A cross-section to the area $s_{ai}$ that is in the recording region on the slice A. Similarly, $R_{bj}$ ($j = 1 \sim m$) is calculated for slice B. Then, for example, $W_{ij}R_{ai}R_{bj}$ was calculated for a connection pair of regions i and j.

Finally, we obtained the connection strength between slice A and slice B regions by adding it between all i and j, $\sum_{i=1}^{n} \left( \sum_{j=1}^{m} W_{ij}R_{ai}R_{bj} \right)$.

*Statistics and reproducibility.* In this study, we generate activity using methods of deep learning. Therefore, there is no opportunity to use statistical tests on that aspect. Tests are used to measure the degree of agreement between the generated activity and the activity of the response. Essentially, we observe the correlation in the scatter plot in two dimensions between the predicted values and the true values, and those values are consistently above 0.5 compared to an expected value of 0, making the significance self-evident. The need to verify significance arose in the context of verifying whether the prediction accuracy represented by that correlation value is related to the relative distance in the brain or the strength of structural wiring. There, we used the Mann–Whitney $U$ test, and for the $p$ values, we applied a Bonferroni correction using the repetition count of 4 in each panel as the sample size when repeatedly calculating the correlation at each stage, to evaluate the significance of the correlation.

## Data availability

The training and prediction data pairs used for spike prediction, as well as the data needed to reproduce the figures, are stored on the same GitHub page while maintaining the appropriate relative path relationships. A non-commercial, academic UGent license applies.

## Code availability

The code used in this study for predicting spike data is available as a zip file named "home_dir_runLSTM.zip" at https://github.com/ShimonoMLab. Running this code will likely generate a learning curve figure similar to Fig. 2 in this paper. Additionally, the code to reproduce Figs. 3–5 of this paper is also available as a zip file named "home_dir_finalplot.zip" on the same GitHub page.

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

## Acknowledgements
MS has been supported by several MEXT Grant-in-Aid for Scientific Research (20H04257, 21H01352, 23K18493) and the Leading Initiative for Excellent Young Researchers (LEADER) program. EN is supported by several MEXT Grant-in-Aid for Scientific Research (B) (22H03661) and Scientific Research (C) (21K02846, 21K12187). The imaging of MRI and immunostaining of this work were performed in the Division for Small Animal MRI, Medical Research Support Center, Graduate School of Medicine, Kyoto University, Japan. We acknowledge Doris Zakian excellent comments to edit this manuscript, and all support from Innovative Support Alliance for Life Science, and the Hakubi Center at Kyoto University to complete this study.

## Author contributions
R.N. erformed the experiments, analyzed the data, wrote the paper. A.S. performed the experiments, analyzed the data. H.T. contributed materials/analysis tools. Kouki Matsuda, Performed the experiments, Contributed materials/analysis tools Eita Nakamura, Analyzed the data, Contributed materials/analysis tools, Wrote the paper Masanori Shimono Conceived and designed the experiments, Performed the experiments, Analyzed the data, Contributed materials/analysis tools, Wrote the paper.

## Competing interests
The authors declare no competing interests.
