## [Peer Review File · Communications Biology]

Reviewers' comments:

Reviewer #1 (Remarks to the Author):

I thank the editor for the opportunity of reviewing this paper on the reproducibility of neuronal interactions after structural disconnections via machine learning. The study has great potential to foster the application of a machine learning framework in exploring biological and possibly functional mechanisms at the neuronal level that are not currently achievable with the classical physiology-based approach. Nevertheless, some concerns must be addressed as the manuscript writing is obscure and accessible to an expert reader.

The first sentence of the abstract is engaging and clearly states the aim of the study. However, the following part reports the results in a compartmental and obscure way for readers blind to the paper's content (i.e. what is a 'new "generating" approach'? How could the generation of activity between two disconnected regions be produced? What are the 'principles in neuroscience' could the generating approach unveil?).

Although the introduction and explanation of the machine learning technique used in the study are necessary, paragraph 1-2 is as vague as skipping to the following improves the reading. The authors should avoid splitting the introduction into sections to maintain the flow and ensure that all the paragraphs are linked. A clear statement of the aim of the study and its hypothesis needs to be included in the introduction.

The discussion is hard to follow. Again, avoiding or reducing the number of sections could make the reading easier.

I have some doubts about point 3-4. Does the author demonstrate in their study that the hierarchy of information processing is related to cell density? This statement could be far-fetched as I could find this evidence trivially reported only in the caption of Figure 3.

Finally, the discussion should be deepened in the meaning of the finding that there is a positive correlation between connection strength and the prediction performance of synchronisation. Is this evidence that severed regions maintain functional connectivity in the absence of structural connectivity? Or that neuronal networks work independently from each other? The authors comment that this is a 'surprising' result. It should be expanded on why.

Reviewer #2 (Remarks to the Author):

The authors trained a machine learning model to predict the future spikes of a population of neurons disconnected from the rest of the brain based on 17 min recordings. They show how a given region's activity can be predicted with high accuracy and how the activity of one region can predict the spontaneous spikes of another disconnected region.

This is an interesting study with novel findings that would help further our understanding of the brain. However, some parts of the paper are unclear and need more details or rewritten; some things might need correction. I would recommend accepting the article if major revisions are done.

Comment:

2.1: There are some discrepancies between the paragraph and the figure. The authors mention the focal loss, but the binary cross entropy is shown in the figure. Also, the sharp loss decrease is before 25 epochs, not 25 to 100.

2.2: I am not sure why the different datasets are named 1 and 2 if they are only used as training and test sets; it is a bit confusing. It makes the reader think that they are pretty much equivalent, but one is later during the recording, they are not independent. I would suggest using a different name that would explain the difference between them more explicitly. (Maybe it's my Python programming coming out, but I think that would help the readers)

"The accuracy of how well the generated time series reproduced the statistical properties of the actual data was evaluated as the average of dataset 1 and dataset 2 " I think there is an issue here. If the goal is to reproduce the properties of the average of something with the training set, the performance evaluation is massively biased. As I understand it here, the performance is calculated on the average between the training set and the validation set; if it is the case, the analysis is not valid. Please, provide a more detailed explanation showing that the training set was not used during the evaluation.

The paragraph starting with "For further evaluation of the synchronization score" is unclear to me, as well as the explanation in the method section. I don't understand why these plots are used and how they help evaluate the performance. It is not a criticism of the method, but I have just never seen this used before, so I think it would help readers like me to have more details on that.

2.4: "However, there was a significant positive correlation between connection strength and predicted sharpness in the first hemisphere (Left hemi.: $p=5.510^{-9}$, Right hemi.: $p=2.010^{-8}$, $p<0.01$, Bonferroni correction), which was common in the left and right hemisphere." This is unclear.

4.2: Why remove the first 30 minutes of recordings? Is it the 20min prescan plus 10 min? In that case, why was 10 min on top of the prescan?

4.3.2: "The dimension of the input and output vectors, which are the number of neurons, was also 128" Why is the input size 128? Earlier, it was said in 4.1. that 1020 electrodes were selected from the 26000 electrode grid. I don't know if there is a link between these different selections, but I think having the rationale for the selection would be helpful so there is no confusion.

4.3.3: "The acceptable delay D was set to 1" What is the time unit of D? Maybe I missed it, but I don't see the unit of time used here; it would help give an idea of the frequency of the spikes, for example. It is also unclear why this would badly interact with the sharpness plot other than just not being an acceptable time delay. Explained like that, it almost looks like a bug in the code that only works with a value of 1.

4.3.5: It is difficult to understand why angles were used instead of actual distances. I don't quite understand how a region at an angle compared to another region is more or less distant.

Reviewer #3 (Remarks to the Author):

The authors propose an interesting manuscript about generating accurate neuronal spikes using the deep artificial neural network LSTM (Long Short Term Memory) to model outcomes based on sequential data. The challenging part in such experiments is not only to predict the firing rates of the neurons but to match the synchronization of firing between different neuron pairs. In this work, the predicted neuronal spikes are tested against their ground truth (the real neural activation) and also investigated in regard to the causes driving the neural activation. The two main aspects the authors evaluate to answer this question are the strength of the (past) structural connections and the physical relative distance of the 16 regions in the mouse brain neocortex that were chosen for this study. By this, the work aims to generate data using an artificial neural network, to explain how much the structural connectivity and the relative distance contribute to a similar pattern of neural activation, as well as to contribute to the reduction of animal experiments by demonstrating that such data can be generated artificially without sacrificing animals. By performing these experiments and analyses, the authors found out that even with disconnected white matter tracts and between distant regions, synthetic spike sequences could be accurately modeled, matching the previously inputted natural neuronal activation patterns not only in terms of firing rate but also in terms of the degree of synchrony. Consequently, the authors drew the conclusion that there must be hidden rules for neuronal activity, producing an activation pattern that the deep learning model was able to detect and

learn. However, unfortunately, no ideas are proposed regarding what could potentially drive these rules; adding this to the discussion would benefit the paper. Overall, this is an intriguing framework with great potential that needs further exploration with a more extensive and variant dataset. Please find some questions and more detailed remarks below.

- 1) Line 49: Perhaps adding the information about how the simultaneous measuring of neurons has developed precisely over the years would help the reader to put the conducted work into perspective and context.
- 2) Line 78: Analyzing neural spike data has not been unexplored for a long time. The first attempts to do so can be traced back to the 1960s. For example, Wilfrid Rall's work, whose mathematical model of neurons used differential equations to describe the electrical activity of neurons over time, has provided important insights and can be seen as a precursor for the construction of an SNN (spiking neural network) model. Similarly, Carver Mead could be cited to give some more background on the more relevant work (e.g., neuromorphic computing that used SNNs to simulate the behavior of biological neurons).
- 3) Are the 16 regions of equal size? Perhaps a bigger visualization (in addition to Figure 1 (e)) of the mouse brain with the 16 divided regions could be helpful. Also, I am asking the size question because if each region has 128 cells (LSTM units) that were selected for the training and testing sequences, what if some smaller regions are overrepresented since the cell number is not normalized with the region's size?
- 4) 2-1. Training process: why did you not use early stop? Overfitting might be a problem; how did you take this into account?
- 5) Regarding the proposal to increase the variability of the dataset: perhaps generating more series of (eventually shorter) sequences and performing k-fold cross-validation, inputting different sequences for every fold (for training) would introduce additional variations in the data and add to the prediction strength, as well as generalizability of the LSTM model. This might also reduce the risk of overfitting the model.
- 6) Figure 5: the bar diagrams c, h, m, r, including the y-axis label, would be great. Perhaps it is also possible to reduce some information or concatenate it.
- 7) Sharpness is often used as an evaluation metric but is explained only in the end. Having a short explanation at the beginning of the results section would be great to help the reader to follow the interesting findings, Figure 3: giving a classification of Theta would be good. Is this the measure of sharpness – what is a good theta in this case?
- 8) Line 425-426: is this not contradictory to the findings reported in the results section (Line 340-343)? You are saying, "no significant correlation was observed between structural connection strength and prediction performance of firing (Line 425), but in the results (Line 340) you are stating that "In general, the results indicate that the prediction performance of the index of firing activity in the multi-layer LSTM is related to both the relative distances between measurement sites and the strength of the structural connections."
- 9) How big is the testing and the training set? 80/20? How many segments were used for training, and how much for testing?
- 10) Line 428: Which hemisphere was it in some cases? Are there any explanations or implementations why that is, in case it is the same hemisphere?
- 11) Line 520: You are mentioning the Transformer model – how would it be different? What benefits in contrast to the LSTM model are there?

The APA citations need to be double-checked because there are oftentimes mistakes. Also, the consistency in the vocabulary could be potentially improved. Different labels for the same concept (e.g. relative distance and angle) make it more confusing and more complicated for the reader to follow.

Referee expertise:

Referee #1: Predictive models and data reduction, cognition, brain disconnections

Referee #2: Machine learning applied to biological samples, brain disconnections

Referee #3: Machine learning applied to neuroimaging, brain disconnections

Reviewers' comments:

Reviewer #1 (Remarks to the Author):

I thank the editor for the opportunity of reviewing this paper on the reproducibility of neuronal interactions after structural disconnections via machine learning. The study has great potential to foster the application of a machine learning framework in exploring biological and

possibly functional mechanisms at the neuronal level that are not currently achievable with the classical physiology-based approach. Nevertheless, some concerns must be addressed as the manuscript writing is obscure and accessible to an expert reader.

Thank you very much for taking the time to review our paper. We truly appreciate your evaluation of our research and the invaluable comments you provided. We have made sincere improvements based on each question, aiming to enhance readability and accuracy. Please take a moment to review the revised manuscript.

The first sentence of the abstract is engaging and clearly states the aim of the study.

However, the following part reports the results in a compartmental and obscure way for readers blind to the paper's content (i.e. what is a 'new "generating" approach'? How could the generation of activity between two disconnected regions be produced?

Thank you for your feedback. Here, the term 'generative' approach specifically refers to a multi-layer LSTM (Long Short-Term Memory) model that learns the rules of activity generation in a certain region and applies this knowledge to generate activity in other regions.

What are the 'principles in neuroscience' could the generating approach unveil?).

Thank you for your feedback. I regret that the expression 'Principle of neuroscience' was too broad. I have now softened it to 'trend governing non-uniformity of cortex'. With this expression, would you find it more understandable?

Although the introduction and explanation of the machine learning technique used in the

study are necessary, paragraph 1-2 is as vague as skipping to the following improves the reading.

To clarify the explanation about the machine learning technique used in this study, we removed some less important sentences and emphasized the motivation for utilizing the multilayer LSTM model.

The authors should avoid splitting the introduction into sections to maintain the flow and ensure that all the paragraphs are linked. A clear statement of the aim of the study and its hypothesis needs to be included in the introduction.

Thank you very much for providing advice that will help improve the readers' understanding. I have removed the separation between sections in the Introduction. Additionally, I have explicitly stated the hypothesis at the end of the Introduction. How does the current wording look to you?

The discussion is hard to follow. Again, avoiding or reducing the number of sections could make the reading easier.

Thank you for your feedback and comments. Based on your suggestions, we have made changes to the manuscript by removing the subsection within 3-5 and incorporating the original 3-2 into section 4, while integrating 3-3 into 3-1, thereby reducing the number of chapters. We would

appreciate it if you could review the current content and let us know if there are any further improvements we can make.

I have some doubts about point 3-4. Does the author demonstrate in their study that the hierarchy of information processing is related to cell density? This statement could be far-fetched as I could find this evidence trivially reported only in the caption of Figure 3.

I apologize for any confusion caused by my previous statement. The content discussed in the original 3-4 (currently 3-3) is a review of past research that differs from our own. The intention was to connect this content to the outlook in the following chapter. To avoid any misunderstanding, we have made edits to the beginning of the original 3-4 (currently 3-3) to clearly indicate that it is not related to the results obtained in this study.

Finally, the discussion should be deepened in the meaning of the finding that there is a positive correlation between connection strength and the prediction performance of synchronisation. Is this evidence that severed regions maintain functional connectivity in the absence of structural connectivity? Or that neuronal networks work independently from each other? The authors comment that this is a 'surprising' result. It should be expanded on why.

Thank you very much for pointing out the issues where the discussion regarding critical points was lacking. Firstly, we have added an explanation as to why the results are deemed 'surprising.' Secondly, we have provided a more detailed expression of the interpretation of these findings.

As a solution to the first issue, we added the following statement: '*Our experiments involved measuring neural activity from brain regions after sectioning them as slices. This means that the connections between those brain regions are severed, and as a result, these regions do not have shared input. Therefore, the factors from outside these two regions that would normally preserve the similarity in neural activity between them are absent.*'

Furthermore, as a solution to the second issue, we incorporated the following passages: '*Such genes and transcription factors are internalized characteristics of each brain region. In other words, these studies indicate that brain regions connected by structural wiring possess similarities in terms of their internal activity generation characteristics. In our research findings, the reason for successful generation between disconnected brain regions is understood to leverage the similarity of activity generation characteristics that each brain region inherently possesses. In essence, connecting the dots, there is a possibility that the similarity of genes and transcription factors is related to the ease of mutual generation of neural activity in each brain region, which we have discovered.*' Additionally, we included the statement: '*To understand the connection between genes, transcription factors, and neural activity characteristics, it is crucial to continuously grasp the multi-layered hierarchy of how proteins produced from genes govern the chemical properties of neurons, such as their structural and chemical channel characteristics, leading to the emergence of electrical properties specific to each cell group.*'

If there are any recent findings or discussions that could be relevant to the current topic beyond what has been discussed, we would greatly appreciate being informed. Thank you very much for your attention and consideration.

Reviewer #2 (Remarks to the Author):

The authors trained a machine learning model to predict the future spikes of a population of neurons disconnected from the rest of the brain based on 17 min recordings. They show how a given region's activity can be predicted with high accuracy and how the activity of one region can predict the spontaneous spikes of another disconnected region.

This is an interesting study with novel findings that would help further our understanding of the brain. However, some parts of the paper are unclear and need more details or rewritten; some things might need correction. I would recommend accepting the article if major revisions are done.

Thank you very much for providing us with numerous valuable suggestions for improvement to accurately convey the intricacies of our data analysis to the readers. This has been an extremely valuable opportunity to consider the perspective of naive readers, which we might not have noticed on our own. We have carefully incorporated the suggested improvements, and we kindly ask you to review the revised manuscript. We sincerely appreciate your understanding of the importance of our research and the thoughtful recommendations for improvement.

Comment:

2.1: There are some discrepancies between the paragraph and the figure. The authors mention the focal loss, but the binary cross entropy is shown in the figure.

I apologize for the fundamental mistake. The vertical axis on the figure represents focal loss, and we have made the necessary corrections.

Also, the sharp loss decrease is before 25 epochs, not 25 to 100.

Thank you for bringing this to our attention. We have made the necessary corrections.

2.2: I am not sure why the different datasets are named 1 and 2 if they are only used as training and test sets; it is a bit confusing. It makes the reader think that they are pretty much equivalent, but one is later during the recording, they are not independent. I would suggest using a different name that would explain the difference between them more explicitly. (Maybe it's my Python programming coming out, but I think that would help the readers)

I apologize for the complexity in the naming. There are two sets of 34-minute time series data (datasets), which were divided into 17-minute segments each to create training and test data. To make this point clearer, we have updated the figures in panels (b) of Figures 3 and 4. I hope this avoids any confusion.

"The accuracy of how well the generated time series reproduced the statistical properties of the actual data was evaluated as the average of dataset 1 and dataset 2 " I think there is an issue here. If the goal is to reproduce the properties of the average of something with the training set, the performance evaluation is massively biased. As I understand it here, the performance is calculated on the average between the training set and the validation set; if it is the case, the analysis is not valid. Please, provide a more detailed explanation showing that the training set was not used during the evaluation.

As I mentioned in the previous response, the distinction between Dataset 1 and Dataset 2 is separate from the differentiation between training and test data. Therefore, during the evaluation of performance, we do not utilize the training data. Thank you for giving this important question.

The paragraph starting with "For further evaluation of the synchronization score" is unclear to me, as well as the explanation in the method section. I don't understand why these plots are used and how they help evaluate the performance. It is not a criticism of the method, but I have just never seen this used before, so I think it would help readers like me to have more details on that.

We added some sentences to explain the intuitive definition and the interpretation of the synchronization score to guide the reader to understand the analysis method. We also added some sentences to explain the definition of sharpness in this section.

2.4: "However, there was a significant positive correlation between connection strength and predicted sharpness in the first hemisphere (Left hemi.: $p=5.510 \cdot 10^{-9}$, Right hemi.: $p=2.010 \cdot 10^{-8}$, $p<0.01$, Bonferroni correction), which was common in the left and right hemisphere." This is unclear.

Indeed, you are absolutely right. Thank you for your comment. It is likely that conducting statistical tests would lead to significant results. However, mixing different metrics for testing could potentially be standardized, but it might make the interpretation less clear. Therefore, we have decided to remove this message.

4.2: Why remove the first 30 minutes of recordings? Is it the 20min prescan plus 10 min? In that case, why was 10 min on top of the prescan?

The reason for excluding the first 30 minutes is that during the observation of raw data, it was confirmed that there are data points where the firing rate is not stable within the initial 30 minutes of the experiment.

The 20-minute pre-scan is a process conducted before the main recording, where only electrodes present in regions where activity can be observed are selected. This is implemented in the application of the device we use to effectively perform the main recording. This point has also been added to Section 4-1 of the main text.

4.3.2: "The dimension of the input and output vectors, which are the number of neurons, was also 128" Why is the input size 128? Earlier, it was said in 4.1. that 1020 electrodes were selected from the 26000 electrode grid. I don't know if there is a link between these different selections, but I think having the rationale for the selection would be helpful so there is no confusion.

I apologize for the oversight. As mentioned in the response to Section 4.2, the pre-scan process efficiently selects electrodes located in the vicinity of active cells. From these electrodes that can measure activity, we extract 128 cells that are present within a region encompassing both the full depth of the neocortex, spanning cortical layers 1 to 6, and is bounded by two lines along the depth direction and the surface and deep faces of the cortex. Further details on this procedure can be found in Shirakami et al., 2021, and a concise explanation of this process, sufficient to avoid any misunderstanding, has been included in this paper for reference.

4.3.3: "The acceptable delay D was set to 1" What is the time unit of D? Maybe I missed it, but I don't see the unit of time used here; it would help give an idea of the frequency of the spikes, for example.

I apologize for the omission. The data is measured at each time step with a bin width of 1 ms, and D is measured using that bin, so the time unit of D is in milliseconds (ms).

It is also unclear why this would badly interact with the sharpness plot other than just not being an acceptable time delay. Explained like that, it almost looks like a bug in the code that only works with a value of 1.

The size of the time window that preserves the causal relationship within the observed time series is determined by the characteristics of the data. In spike data, it is well-known that long-term correlations are effective, but short-term causality also plays a significant role. Increasing the time bin might improve performance as it widens the 'allowable range,' but for instance, if we expand a 1ms window to 2ms, it might combine two interactions that occurred separately within each 1ms window. This could lead to the two interactions being treated as if they had a fixed relationship, even if they actually switched based on past context, potentially reducing performance. This is just one mechanism, but at least when increasing the time bin size, a decrease in performance is possible as a phenomenon.

In this study, the adjustment of the window size was a process aimed at improving the performance of predictions, and delving into the mechanism of how the window size affects the

performance is slightly beyond the focus of this research. However, it is an extremely interesting point, and we would like to investigate and validate it in future studies. Thank you for providing your comments.

4.3.5: It is difficult to understand why angles were used instead of actual distances. I don't quite understand how a region at an angle compared to another region is more or less distant.

We apologize for the confusion caused by our inconsistent use of terminology. Thank you for clarifying the issue. In the paper, terms such as distance, relative distance, relative angle, and angle were referring to the same quantity. Therefore, we have unified all these terms as "relative angle."

Essentially, we classified the regions based on angles, and having a wide relative angle implies a greater relative distance. As the cortex is approximately spherical, we took cross-sections along directions perpendicular to the cortical surface and grouped them systematically based on angles. Using angles for grouping made the classification more straightforward.

By unifying the terminology, we aim to make the concepts easier to understand. If you have any further questions, please feel free to ask.

Reviewer #3 (Remarks to the Author):

The authors propose an interesting manuscript about generating accurate neuronal spikes using the deep artificial neural network LSTM (Long Short Term Memory) to model outcomes based on sequential data. The challenging part in such experiments is not only to predict the firing rates of the neurons but to match the synchronization of firing between different neuron pairs. In this work, the predicted neuronal spikes are tested against their ground truth (the real neural activation) and also investigated in regard to the causes driving the neural activation. The two main aspects the authors evaluate to answer this question are the strength of the (past) structural connections and the physical relative distance of the 16 regions in the mouse brain neocortex that were chosen for this study. By this, the work aims to generate data using an artificial neural network, to explain how much the structural connectivity and the relative distance contribute to a similar pattern of neural activation, as well as to contribute to the reduction of animal experiments by demonstrating that such data can be generated artificially without sacrificing animals. By performing these experiments and analyses, the authors found out that even with disconnected white matter tracts and between distant regions, synthetic spike sequences could be accurately modeled, matching the previously inputted natural neuronal activation patterns not only in terms of firing rate but also in terms of the degree of synchrony. Consequently, the authors drew the conclusion that there must be hidden rules for neuronal activity, producing an activation pattern that the deep learning model was able to detect and learn.

We sincerely appreciate the time and effort you have dedicated to thoroughly understanding our research. Furthermore, we are grateful for your valuable insights on this intriguing paper. It brings us great joy to see that you have gained a comprehensive understanding of our study. Below, we have provided sincere responses to your inquiries. Could you take a moment to review them?

Once again, thank you very much for your thoughtful feedback and consideration.

However, unfortunately, no ideas are proposed regarding what could potentially drive these rules; adding this to the discussion would benefit the paper.

This aspect is indeed a highly challenging issue. We realize that using terms like 'principles in neuroscience' in the abstract might have been problematic, and as such, we have made the necessary revisions.

However, as pointed out by the reviewers, the observed characteristic of 'mutual generation of activity between disconnected brain regions' in our study is highly suggestive and could serve as a catalyst for further investigations into novel principles in neuroscience. Taking this into consideration, I have included the following statement in the discussion section of the paper.:

'genes and transcription factors, known to be related with structural wiring pattern, are internalized characteristics of each brain region. In other words, these studies indicate that brain regions connected by structural wiring possess similarities in terms of their internal activity generation characteristics. In our research findings, the reason for successful generation between disconnected brain regions is understood to leverage the similarity of activity generation

characteristics that each brain region inherently possesses. In essence, connecting the dots, there is a possibility that the similarity of genes and transcription factors is related to the ease of mutual generation of neural activity in each brain region, which we have discovered.' Additionally, we included the statement: '*To understand the connection between genes, transcription factors, and neural activity characteristics, it is crucial to continuously grasp the multi-layered hierarchy of how proteins produced from genes govern the chemical properties of neurons, such as their structural and chemical channel characteristics, leading to the emergence of electrical properties specific to each cell group.'*

It's possible that the points we mentioned here may not be what the reviewers were expecting in terms of the discussion. If there are other important aspects or topics that should be addressed at this point, please feel free to provide further guidance, and we will be sure to address them accordingly.

Overall, this is an intriguing framework with great potential that needs further exploration with a more extensive and variant dataset. Please find some questions and more detailed remarks below.

Thank you for taking the time to thoroughly understand our research and provide your valuable evaluation and feedback. We will keep to sincerely respond to your comments.

1) Line 49: Perhaps adding the information about how the simultaneous measuring of neurons has developed precisely over the years would help the reader to put the conducted work into

perspective and context.

Thank you for your important advice. We improved the literature as follows:

The ability to simultaneously record from multiple neurons has significantly improved over the years, thanks to advances in electrode technologies. From the advent of transistor computers and microelectrode probes in the 1950s, there has been a remarkable trend where the number of neurons that can be monitored simultaneously has approximately doubled every seven years [Stevenson and Kording, 2011]. Today, with the advent of novel electrode technologies, we can record activity from hundreds, thousands, or even tens of thousands of neurons at the same time [Hong and Lieber, 2019; Nurmikko, 2020; Urai et al., 2022]. These advances have been fueled in part by improvements in the scalability and accessibility of input/output interfaces, the reduction of electrode sizes to afford a higher density and sampling resolution, and the design of macroporous structures to increase the sampling volume without causing substantial damage to neural tissues [Hong and Lieber, 2019; Nurmikko, 2020].

And, we cited these new reports.

Hong, G., & Lieber, C. M. (2019). Novel electrode technologies for neural recordings. *Nature Reviews Neuroscience*, 20(6), 330-345.

Nurmikko, A. (2020). Challenges for large-scale cortical interfaces. *Neuron*, 108(2), 259-269.

Urai, A. E., Doiron, B., Leifer, A. M., & Churchland, A. K. (2022). Large-scale neural recordings call for new insights to link brain and behavior. *Nature neuroscience*, 25(1), 11-19.

2) Line 78: Analyzing neural spike data has not been unexplored for a long time. The first attempts to do so can be traced back to the 1960s. For example, Wilfrid Rall's work, whose mathematical model of neurons used differential equations to describe the electrical activity of neurons over time, has provided important insights and can be seen as a precursor for the construction of an SNN (spiking neural network) model. Similarly, Carver Mead could be cited to give some more background on the more relevant work (e.g., neuromorphic computing that used SNNs to simulate the behavior of biological neurons).

Thank you for suggesting these past important literatures. We edited the part as follows: "Given

the similarities between artificial neural networks (ANNs) and motifs in the nervous system, it might be expected that ANNs could be applied to generate spike data with properties resembling real-world neural activity. There were attempts to analyze neural spike data can be traced back to as early as the 1960s, with seminal work by pioneers such as Wilfrid Rall, whose mathematical models used differential equations to describe the temporal dynamics of neuronal electrical activity [Gers et al. 2002]. Rall's work was essential in laying the groundwork for what would eventually become spiking neural network (SNN) models. Another notable contributor to this field is Carver Mead, who made significant strides in neuromorphic computing, using SNNs to simulate the behavior of biological neurons. Despite these historical precedents, the full potential of ANNs for generating neural spike data has not yet been fully realized and warrants further exploration.”

If there are any more problems, please feel free to ask us again.

3) Are the 16 regions of equal size? Perhaps a bigger visualization (in addition to Figure 1 (e)) of the mouse brain with the 16 divided regions could be helpful. Also, I am asking the size question because if each region has 128 cells (LSTM units) that were selected for the training and testing sequences, what if some smaller regions are overrepresented since the cell number is not normalized with the region's size?

For each of the 16 regions, we selected the same 128 cells within the measurement area. As a result, the cell density varies, leading to differences in spatial size that are inversely proportional to cell density. However, some form of normalization is necessary for the analysis, and the selection

method we used can be considered a natural approach among possible choices. In future studies, we hope to conduct analyses where the spatial size is fixed, but the number of cells varies for each region. We also discussed similar points in the discussion section, and we have added the above-mentioned aspect as a future task. It is also important to understand that the improvement in accuracy obtained here is not reflected in the loss function.

4) 2-1. Training process: why did you not use early stop? Overfitting might be a problem; how did you take this into account?

It seems that our wording might have caused some confusion. In the main text, we stated, "This decision was based on the observation that even though the value of the loss function had converged at around 25-100 epochs, the precision of the firing rate and the reproducibility of synchronous firing continued to improve up to 350 epochs." Therefore, we stopped training at 350 epochs. However, in the figure, it was still written as "decreases sharply at 25-100 epochs," which may have given the impression that we should have stopped training even earlier. To avoid this confusion, we have now revised the wording in the figure to align with the main text.

5) Regarding the proposal to increase the variability of the dataset: perhaps generating more series of (eventually shorter) sequences and performing k-fold cross-validation, inputting different sequences for every fold (for training) would introduce additional variations in the data and add to the prediction strength, as well as generalizability the of the LSTM model.

This might also reduce the risk of overfitting the model.

I apologize for the insufficient explanation regarding the choice of stopping epoch numbers in the previous response, which led to the need for this question. In addition to the previous problem relating to this issue, there are two main reasons why we did not perform k-fold cross-validation:

(1) In Deep Neural Networks (DNNs), k-fold cross-validation is typically used when there is limited data available. However, in our case, we have maximized the utilization of data frames, and the data size has already reached the memory limit of the server.

(2) The current approach has already achieved non-trivial predictions, and there are numerous methods available to further improve performance.

Considering the above situation, we obtained meaningful results with the test data after training on the training data. While there is potential for performance improvement by fine-tuning to address overfitting in more detail, the current analysis, which involves stopping calculations at a certain combination of epoch numbers while utilizing all available data within the limited memory constraints, can still be considered a good approach.

Moving forward, we plan to improve by optimizing the computational environment and exploring the introduction of Transformer and other novel techniques. These areas represent challenges for improvement. We hope that this explanation provides a clear understanding of our approach and decisions. Thank you for your understanding.

6) Figure 5: the bar diagrams c, h, m, r, including the y-axis label, would be great. Perhaps it is also possible to reduce some information or concatenate it.

Thank you for your feedback. We have added labels to each figure and increased the font size to improve readability. Additionally, we have unified the terminology to 'relative angle,' consistent with the expressions used in the text.

7) Sharpness is often used as an evaluation metric but is explained only in the end. Having a short explanation at the beginning of the results section would be great to help the reader to follow the interesting findings, Figure 3: giving a classification of Theta would be good. Is this the measure of sharpness – what is a good theta in this case?

Thank you for bringing this issue to our attention and helping to make it more understandable for the readers. We added some sentences to explain the intuitive definition and the interpretation of the synchronization score to guide the reader to understand the analysis method. We also added some sentences to explain the definition of sharpness in this section.

8) Line 425-426: is this not contradictory to the findings reported in the results section (Line 340-343)? You are saying, “no significant correlation was observed between structural connection strength and prediction performance of firing (Line 425), but in the results (Line 340) you are stating that “In general, the results indicate that the prediction performance of the index of firing activity in the multi-layer LSTM is related to both the relative distances between measurement sites and the strength of the structural connections.”

I apologize for a crucial mistake that occurred. The statement written in Line 340 was incorrect.

The accurate statement is 'the prediction performance of synchronization is related to both relative angles and the strength of the structural connections.' I have made the necessary correction.

9) How big is the testing and the training set? 80/20? How many segments were used for training, and how much for testing?

Both the training data and the test data are 17 minutes long, making the split 50/50.

10) Line 428: Which hemisphere was it in some cases? Are there any explanations or implementations why that is, in case it is the same hemisphere?

In predicting synchrony, the first quadrant mainly consists of excitatory neurons, which makes it more likely for synchrony between excitatory neurons in either hemisphere to occur compared to predictions involving inhibitory interactions in other quadrants. This could be considered a natural phenomenon.

11) Line 520: You are mentioning the Transformer model – how would it be different? What benefits in contrast to the LSTM model are there?

We revised the sentences to explain the feature of the Transformer model and how this model could be beneficial for improving the data generation.

The APA citations need to be double-checked because there are oftentimes mistakes. Also, the consistency in the vocabulary could be potentially improved. Different labels for the same concept (e.g. relative distance and angle) make it more confusing and more complicated for the reader to follow.

Thank you for bringing this issue to my attention. I have carefully organized the formatting of the reference

list. Please review it below.

REVIEWERS' COMMENTS:

Reviewer #1 (Remarks to the Author):

The authors addressed my concerns for the most part. Although the editing of the introduction was limited to the switching of the paragraphs, I find the explanation of the model much more straightforward now. The improvements made to the discussion enriched the section and highlighted the impact of the findings.

Reviewer #2 (Remarks to the Author):

Thank you for answering my questions and clarifying some of the explanations.

The number of epochs for the sharp decrease has not been modified in the second figure's legend in the text, but I see the figure changed at the end of the file. Just make sure it is updated in the final file.

It would have been helpful to highlight the changes in the text as it made it much longer to track the changes without it.

Besides that, I think the authors answered my concerns and questions.

Reviewer #3 (Remarks to the Author):

Dear Editor,

all my comments and suggestions have been addressed sufficiently. The authors took every comment into consideration, provided an answer, and edited their manuscript accordingly.

The issues raised by me as a reviewer have been solved. I believe this is a good paper that can be published once also the other reviewers agree.